# Clinical Application of Diffusion Tensor Imaging for a Brachial Plexus Injury

**DOI:** 10.3390/diagnostics12071687

**Published:** 2022-07-11

**Authors:** Giulio Vara, Gianmarco Tuzzato, Giuseppe Bianchi, Marco Miceli, Luca Spinardi, Rita Golfieri, Raffaella Rinaldi, Giancarlo Facchini

**Affiliations:** 1Diagnostic and Interventional Radology, IRCCS Istituto Ortopedico Rizzoli, Via Pupilli 1, 40136 Bologna, Italy; marco.miceli@ior.it (M.M.); raffaella.rinaldi@ior.it (R.R.); giancarlo.facchini@ior.it (G.F.); 2Department of Radiology, IRCSS Azienda Ospedaliero Universitaria di Bologna, 40138 Bologna, Italy; rita.golfieri@unibo.it; 3Unit of 3rd Orthopedic and Traumatologic Clinic Prevalently Oncologic, IRCCS Istituto Ortopedico Rizzoli, Via Pupilli 1, 40136 Bologna, Italy; gianmarco.tuzzato@ior.it (G.T.); giuseppe.bianchi@ior.it (G.B.); 4Diagnostic and Interventional Neuroradiology Unit, S. Orsola–Malpighi Hospital, 40138 Bologna, Italy; luca.spinardi@aosp.bo.it

**Keywords:** brachial plexus, tractography, diffusion tensor imaging, radiculopathy, multishell

## Abstract

Brachial plexus injuries are commonly diagnosed clinically, as conventional imaging has a low sensitivity. In recent years, diffusion tensor imaging has established a clinical role in the study of the central nervous system and, while still presenting some limitations due to the technical complexity of the acquisition method, is showing promising results when applied to peripheral nerves. Moreover, deterministic fiber tracking with the Euler’s method and multishell acquisition are two novel advances in the field which contribute to enhancing the reliability of the technique reducing the respiratory and inhomogeneity artifacts in this “magnetically complex” region, and better isolating the fibers in a heterogeneous territory. Here, we report a case of brachial plexus traumatic injury, a healthy reference subject, and details on the acquisition protocol of the reconstruction algorithm.

A 57-year-old man was admitted to the emergency department after a motorbike accident. A CT scan was ordered, demonstrating a mandibular fracture and a subarachnoid hemorrhage. The fracture was treated surgically, and the subarachnoid hemorrhage conservatively. After the recovery, he reported a palsy of the right arm, and an MRI of the shoulder and the cervical spine was ordered, performed 4 weeks after the accident on a 3T scanner. In agreement and with the consent of the patient, a FIESTA, diffusion tensor imaging (DTI) and a 3D short-tau inversion recovery sequences were added to the protocol. The study showed an increased signal of the right brachial plexus on the STIR and the T2-weighted trace of the DTI, predominantly at the trunk and cord of C5; moreover, STIR hyperintensity and T2-weighted trace isointensity were noted on the rotator cuff muscle ipsilateral to the lesion (Figure 1). An offline reconstruction of the nerve fibers was performed with DSI Studio, confirming the asymmetric enlargement of the trunks and cords of the right brachial plexus, and clearly depicting the continuity of the fibers in the trunk of C5, with a significative decrease in local normalized quantitative anisotropy (Figure 2). The examination of the cervical spine was unremarkable; no sufferance of the spinal cord or pseudomemyngoceles were noted. The findings concurred to diagnose axonotmesis of the trunk of C5 (grade III lesion), which was confirmed by an immediately consequent electromyogram (Figure 3). The patient was treated conservatively and is currently recovering.

Among imaging studies, MRI is the modality of choice to study brachial plexus pathologies, but it shows a poor sensitivity, as low as 60%, when assessing post-ganglionic injuries [1]. A concurrence of findings can guide the diagnostic process of the radiologist, provided by different sequences:FIESTA: This steady-state sequence is not very susceptible to flow artifact, so it provides an excellent signal-to-background contrast for nerve roots and allows the detection of pseudomyelomeningoceles. Its fast acquisition time, <2 min, allows it to be incorporated in a standard acquisition protocol for the spine, or even implemented in emergency departments with an available MRI [1]. It has substituted the previous gold standard, the CT-myelography, which has the disadvantages of contrast media administration in the spinal canal and the use of ionizing radiations (particularly important in the obstetric lesions) [2].STIR: Nerves are isointense compared to muscle in T2-weighted sequences, and an hyperintensity can be a consequence of various pathological processes. The presence of a nodular formation on the course of the nerve is indicative of a neuroma, the detection of which MRI has shown a great accuracy of up to 99% [1,3]. The STIR sequence is often preferred over the T2 with fat suppression due to the homogenous fat saturation obtained. Moreover, the depiction of nerve fibers can be further enhanced with the administration of gadolinium, which suppresses the signal generated by the vessels improving the signal-to-background of the nerves [3]. This sequence also allows examining for muscular signal alteration, to properly stage the nerve injury [4].DWI/T2-weighted trace: DWI significatively improves the conspicuity of the nerves compared to T1- and T2-weighted sequences, and is the best alternative to DTI, over which it has the advantages of lower acquisition time and limited need of post-processing [5,6]. The most used b values are 600–700 (so that b × ADC = 1) [7]. A lower *b*-value with a reduced number of excitations can be added to the sequence, to calculate the ADC map, particularly useful in oncological patients. T2-weighted trace is the isotropic DWI resulting from the DTI post-processing, today a task often performed automatically by the machine’s console, so every study comprising a DTI sequence has isotropic DWI images available [8].DTI: Pioneered on the peripheral nervous system in the last decade, the DTI sequences provide the data to perform the nerve fiber tractography, clearly depicting the anatomy of the nerves and allowing to calculate novel in vivo biomarkers, such as fractional and more recently quantitative anisotropy [9,10]. Quantitative anisotropy is less susceptible to edema surrounding the fibers, so a decrease in fractional anisotropy alone should be regarded as edema, while a reduction in quantitative anisotropy is necessary to confirm a structural change [11]. In recent years, vendors have provided radiologists with proprietary applications to perform in-line post-processing, even if advanced studies are possible only on third-party software. Distortions caused by field inhomogeneity can be addressed with post-processing correction available in most software, with the optional addition of an inversed-encoding b0 sequence (“blip-in-blip-out” method) [12]. Manufacturers are addressing this topic by developing coils with a dedicated geometry [3]. A detailed visualization of the nerve fibers is extremely helpful in surgical planning, giving more information when choosing between nerve graft or a nerve transfer [13]. In the last few years, the multishell technique has emerged, and it consists of acquiring at least two DTI sequences with different *b*-values, increasing the accuracy of the tractography and reducing the in-scanner motion artifacts [14]. The multishell technique is being tested on the central nervous system, and very sparse applications on peripheral nerves have been published [15,16]. In this paper, the authors present a first attempt of multishell acquisition of the brachial plexus performed on a healthy volunteer (Figure 4). Diffusion signal in vivo originates from water represented in different structures, not only neurons but free water and glial cells as well. Conventional DTI-based fiber tracking results in a hampered representation of the tracks when they run in proximity to other water molecules with such different orientation modules, thus not matching the prediction of the Stejskal–Tanner equation:$$\frac{s}{s_{0}}=e^{-bD}$$
*S* = signal; *b* = *b*-value; *D* = Diffusion constant.


Measuring of signals at multiple *b* values helps identify “subpopulations” of spins by observing how the signal differs from the prediction of the Stejskal–Tanner equation [17].

Technical details on the sequences used are reported in Table 1.

Tractography can be performed with several pieces of open-source software; the presented cases have been elaborated with DSI Studio [18]. Through an accessible graphical user interface, it allows generating tracts of the whole volume of study or starting from a region of interest. Several different strategies are adopted when selecting the optimal starting point; in these cases, they were placed on the foramina on the sagittal plane, as post-ganglionic lesions were suspected.

For the presented tractographies, the diffusion data were reconstructed using generalized q-sampling imaging with a diffusion-sampling length ratio of 1.25 [19]. Q-sampling imaging was the selected reconstruction method, with it being not bound to a model and less sensitive to edema [11]. The anisotropy threshold was 0.011. The angular threshold was 70 degrees, adding some tolerance to the reported ex vivo studies of the rootlets of the brachial plexus [20]. The step size was 0.01 mm. Tracks with length shorter than 10 or longer than 1200 mm were discarded. A total of 1,000,000 tracts were calculated.

This case report shows the potential of DTI imaging for traumatic peripheral nerve lesions, which could help identifying optimal candidates for surgery while providing otherwise unimaginable anatomic detail to the surgeon [21]. Moreover, in vivo biomarkers of nerve functionality have numerous applications. While acquisition time using the newest techniques may be long, the addition of these should be considered thoroughly when they can provide useful clinical information [22].

## Figures and Tables

**Figure 1 diagnostics-12-01687-f001:**
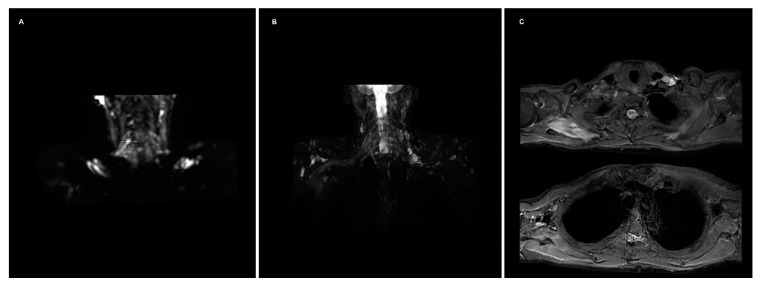
(**A**) T2-weighted trace depicts an increased diffusion restriction on the right brachial plexus; the depiction of the chords on the maximum intensity projection reconstruction is impaired by inhomogeneity artifacts at the lung apices. (**B**) 3D STIR sequence shows hyperintensity of the right plexus, predominant at C5, and of the rotator cuff muscles; conspicuity of the roots and trunks is partially decreased by the presence of multiple lymph nodes. (**C**) Axial STIR images depicting the hyperintensity of the rotator cuff muscles, more evident on the subscapularis.

**Figure 2 diagnostics-12-01687-f002:**
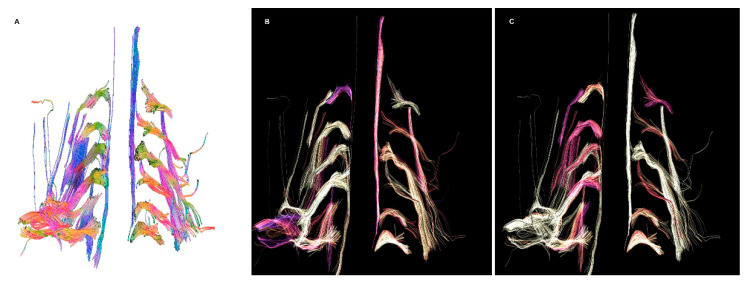
(**A**) Fiber tracking showing an enlargement of the right brachial plexus, more marked in C5. (**B**) No significative differences in fractional anisotropy were noted, besides the cords of the right brachial plexus. (**C**) Decreased normalized quantitative anisotropy of the root and trunk of C5, and less evident of C6.

**Figure 3 diagnostics-12-01687-f003:**
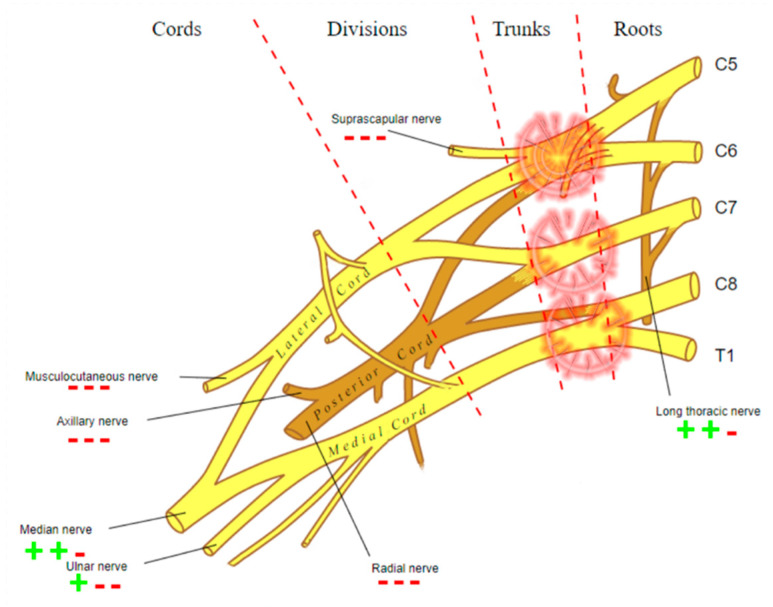
Graphic representation of the damage of the brachial plexus detected by electromyogram.

**Figure 4 diagnostics-12-01687-f004:**
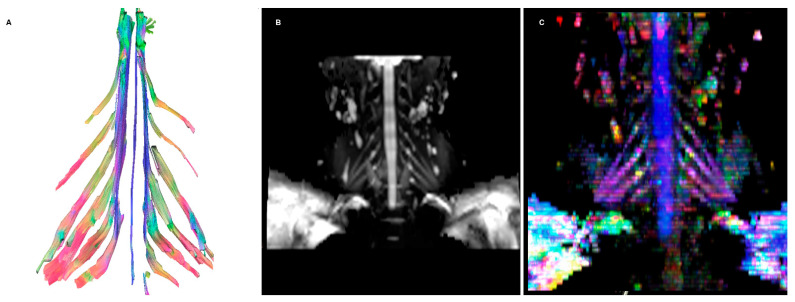
(**A**) Fiber tracking showing symmetrical nerve roots, with significantly less noise compared to the regular DTI acquisition. (**B**) DWI isotropic images resulting from the acquisition. (**C**) Colored orientation distribution function (ODF) map.

**Table 1 diagnostics-12-01687-t001:** Described sequences’ acquisition parameters on a 3T scanner.

Sequence	FOV	Voxel Size (mm)	TE/TR (ms)	*b* Value (s/mm^2^)	Acquisition Time (mm:ss)
Fiesta	18	0.6 × 0.7 × 2	1.5/5.6		2:58
3D STIR	42	1.5 × 1.5 × 1.5	120/2800		4:19
DWI	40	4.0 × 4.2 × 3.6	Min/5099	625	3:41
DTI	36	3.6 × 3.6 × 3.6	Min/7792	800 (30 dir.)	4:19
Multishell DTI	36	3.6 × 3.6 × 3.6	Min/7792	300–600–1000	14:51

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
