# Peer review of "Clinical Application of Diffusion Tensor Imaging for a Brachial Plexus Injury"

_diagnostics, 2022, doi:10.3390/diagnostics12071687_

Round 1

Reviewer 1 Report

Thank you very much for submitting this manuscript.

I have a few comments to improve the manuscript:

1. Fig 1: Images are very small. Most of the images are signal void air. Please magnify the pictures so the readers can see the pathology.

2. The text after the table is a bit disorganized: 

   A. The sentence starting with: "Several different are ..." is unclear. Likely something has been deleted from this sentence. Please rewrite it.

   B. The paragraph starting with " The diffusion data were..." is vague. It is unclear if you are talking about your protocol or mentioning the previous studies.

I wish we had the chance to see the Multi-Shell DTI from the same patient rather than a normal volunteer. If you work in a busy orthopedic hospital, you will see patients with brachial plexus injuries every day. If you can add another patient with brachial plexus injury on multi-shell DTI, your paper would be much more interesting.

3. Please consider adding a new paragraph/table about the physics of Multi-shell DTI, how it is different than traditional DTI, its advantages and disadvantages compared to traditional DTI.

4. References are relatively old. Please consider repeating the review literature. 

Thank you very much

Author Response

Dear Reviewer, 

Thank you very much for the time invested in carefully reviewing this manuscript.

As far as the picture size is concerned, I have already arranged with the editor to upload larger images with a higher resolution in the eventual proofing phase.

The text was indeed misleading and incomplete were you pointed, thank you very much for the attention and sincere apologies for the mistakes. The sentences have been corrected.

A short paragraph on the physical advantages of the multishell technique has been added.

Bibliography now includes more recent studies; however, older studies have been included as they were considered of historical importance.

Thank you very much for your time.

Reviewer 2 Report

Work sensible, factual, written in scientific language please be admitted to next stages 

Author Response

Thank you very much for the comments, and for expending your time reviewing this manuscript.